# Digital Innovative Governance of the Indonesian Creative Economy: A Governmental Perspective

**Dina Dellyana** [1,*], **Nina Arina** [1] **and Tribowo Rachmat Fauzan** [2]

1   School of Business and Management, Institut Teknologi Bandung, Bandung 40251, Indonesia
2   Logistic Business Study Program, Department of Business Administration,
    Faculty of Social and Political Sciences, Universitas Padjadjaran, Sumedang 45363, Indonesia;
    tribowo.fauzan@unpad.ac.id
*   Correspondence: dina.dellyana@sbm-itb.ac.id

**Abstract:** The digital transformation of the creative economy has presented Indonesia with a unique set of challenges and opportunities, demanding innovative governance strategies to navigate this dynamic landscape. This research focuses on the nuanced governance mechanisms employed by the Indonesian government to foster, regulate, and harness the full potential of its digital creative economy. Utilizing a qualitative methodology, this study investigates the experiences, perspectives, and actions of key governmental actors, policymakers, and stakeholders. Semi-structured interviews and document analysis reveal the intricate interplay between government policies and civil servants in Indonesia that takes place in order to manage the creative economy in this developing country. The findings shed light on the adaptive strategies and policies implicated in the creative economy, providing insights into the understanding and collaboration between civil servants that can inform not only Indonesia, but also other nations seeking to harness the transformative power of this rapidly evolving sector. This research contributes to a deeper understanding of the complex relationship between governance and the creative economy, highlighting the need for a digital innovation co-creation scheme by which civil servants can navigate the digital age of creative industries.

**Keywords:** civil servants; creative economy; digital innovation; governance

## 1. Introduction

The creative economy encompasses a diverse range of sectors, including digital content production, software development, gaming, design, and various forms of artistry [1,2]. These sectors are characterized by their propensity to generate high-value intellectual property, foster entrepreneurship, and enhance cultural and creative expression [3,4]. The Indonesian government, recognizing the transformative power of the creative economy, has undertaken various initiatives to stimulate growth in this domain [5]. From crafting favorable policies and incentives to nurturing talent and innovation, Indonesia's efforts in the realm of digital creativity have garnered both domestic and international attention [6].

However, navigating the intricacies of this evolving landscape presents multifaceted challenges for the Indonesian government. The delicate balance between fostering creativity and protecting intellectual property, ensuring equitable access to opportunities, promoting sustainability, and addressing regulatory concerns remains a formidable task [7]. To shed light on these complexities, this research project adopts a governmental perspective, seeking to understand how Indonesian policymakers are adapting to the demands and opportunities of the digital creative economy. It aims to provide insights into the innovative governance strategies, policies, and practices that have been employed to promote sustainable growth while addressing regulatory and societal concerns.

While the study of the creative economy and its governance has gained increasing attention in recent years, there is a notable research gap with regard to the specific governmental perspective on this matter, particularly in the Indonesian context [8]. The existing



literature predominantly focuses on either broader global trends and challenges in the creative economy or, to a limited extent, on the role of the private sector and entrepreneurs in driving innovation within this sector [9,10]. However, there is a distinct dearth of comprehensive research that systematically examines the strategies, policies, and challenges faced by the Indonesian government in fostering and regulating its creative economy [11].

Moreover, while some research has explored governmental efforts to promote innovation and entrepreneurship in other sectors, such as technology and manufacturing [12,13], there is a unique set of challenges and opportunities specific to the creative economy. These challenges encompass issues related to intellectual property rights, talent development, cultural preservation, and the convergence of traditional and creative industries [14]. Understanding how the Indonesian government navigates these complexities and adapts its governance strategies is crucial not only for policymakers in Indonesia but also for other emerging economies seeking to leverage the creative potential of their creative sectors [5,15].

Furthermore, the rapidly evolving nature of the creative economy calls for a nuanced analysis of innovative governance mechanisms [16]. With the advent of new technologies, platforms, and business models, the strategies employed by governments need to be adaptive and forward-looking [17]. Exploring the dynamic nature of governance in this context is essential for developing actionable insights that can inform policy and practice [18]. Therefore, this research project aims to bridge this significant research gap by delving into the governmental perspective of the Indonesian creative economy, providing a comprehensive understanding of the challenges, innovations, and best practices that can guide both Indonesia and other nations on their journey toward harnessing the full potential of this transformative sector.

This research endeavor is not only pertinent for policymakers and stakeholders within Indonesia, but also holds broader significance for nations seeking to harness the transformative potential of their own digital creative economies. By examining the Indonesian case, we aim to draw lessons and inspiration for innovative governance that can be applied in various global contexts, ultimately contributing to a richer understanding of the intricate relationship between governments and the burgeoning creative economy. As digital technologies continue to reshape economic structures and consumer behaviors, governments worldwide are confronted with the imperative task of formulating innovative governance strategies to harness the full potential of this sector.

### 1.1. Creative Economy and Global Trends

The creative economy has evolved into a significant economic force over the past few decades. Scholars such as Howkins (2002) and Markusen et al. (2008) have contributed seminal works that conceptualize the creative economy, emphasizing its role in driving innovation, generating employment, and enhancing the overall quality of life [19,20]. The digital creative economy also emerged in the last few decades, which has witnessed exponential growth, transforming the global economic landscape [21]. Studies have highlighted the emergence of new sectors, including digital content production, animation, gaming, and design, driven by technological advancements and changing consumer behaviors [22,23]. These trends underline the significance of understanding how governments engage with and govern this evolving economic domain [16]. One of the central arguments in the literature is the creative economy's potential to foster economic growth [24]. Researchers have explored the relationship between creative industries and regional development, emphasizing the capacity of creative activities to generate jobs, stimulate tourism, and enhance the overall quality of life in urban and rural areas [25,26]. The creative economy's role as a driver of economic diversification has also been explored, particularly in the context of post-industrial societies [27].

The creative economy is profoundly influenced by globalization trends. In a globalized world, creative products and services can reach audiences far beyond national borders [28]. As a result, scholars have examined the impact of globalization on creative industries, discussing issues such as cultural diversity, transnational flows of creative goods, and

the challenges of protecting intellectual property rights in a global context [28,29]. The globalization of creative markets has raised questions about the balance between cultural homogenization and cultural diversity [30]. Advancements in technology, particularly the Internet and digital technologies, have profoundly shaped the creative economy [12]. Scholars have discussed the democratizing effects of digital platforms, which have lowered barriers to entry for creative entrepreneurs and transformed distribution models [31,32]. The rise of the gig economy and the freelance workforce within the creative sector also exemplify the impact of technology on labor and production in creative industries [33].

### 1.2. Governmental Role in Fostering Innovation

Innovation, often defined as the introduction of new ideas, products, processes, or technologies, is a primary driver of economic growth [34]. Researchers have emphasized the role of innovation in stimulating productivity, creating jobs, and enhancing the competitiveness of nations [35,36]. This foundational understanding underscores the significance of government intervention in facilitating innovation [37]. Governments worldwide have formulated policies and strategies to support innovation across various sectors [38]. Lundvall (1992) introduced the concept of "national systems of innovation", highlighting the interconnectedness of firms, universities, research institutions, and government agencies in fostering innovation [39]. Studies scholars have further detailed how government policies, including research funding, intellectual property protection, and regulatory frameworks, influence innovation processes. Innovation policies are often shaped by regional and global considerations [40,41]. Rodrik (2011) argues that the "trilemma of globalization" poses challenges for governments, as they must balance the demands of globalization, democracy, and national sovereignty [42]. Innovation policies are influenced by international agreements, trade policies, and global supply chain dynamics [43]. Researchers also have explored the role of cities in driving innovation and fostering vibrant creative economies [44,45].

Governments worldwide recognize the potential of the digital creative economy to drive economic growth and innovation [46]. A substantial body of the literature emphasizes the pivotal role of governments in innovation ecosystems. This role includes setting policies, creating infrastructure, and fostering collaboration between the public and private sectors [47,48]. Governments are not only policy enablers but also key actors in innovation ecosystems, shaping the direction of technological progress and economic development [49]. As innovation continues to be a driving force in the global economy, understanding the nuanced ways in which governments can promote and facilitate innovation remains a critical area of research and policymaking. However, the unique characteristics of the digital creative economy require tailored governmental strategies [50].

### 1.3. Indonesian Creative Economy Landscape

The Creative Economy Plan in Indonesia is primarily influenced by the concept of creative industries in the UK. This definition places emphasis on innovation and intellectual property, specifically referring to the "innovative" creative industries. Additionally, it encompasses the inclusion of "traditional cultural industries" as identified by Fahmi, Koster, and van Dijk in 2016 [5]. Indonesia has consistently advocated for the development and promotion of the creative economy. In 2018, Indonesia provided support for the Bali Agenda for Creative Economy, and also witnessed the United Nations General Assembly's approval of a resolution designating 2021 as the Year of the Creative Economy for Sustainable Development. As the designated host country for the G20 summit in 2022 and the chair of the ASEAN secretariat in 2023, our objective is to enhance coordination and collaboration at the local, regional, and global levels to bolster the creative economy as a catalyst for content-oriented economic growth. The creative economy serves as a valuable resource for facilitating the availability of employment opportunities, particularly within micro-, small-, and medium-sized firms, during periods characterized by swift transformations in labor markets. The creative economy holds significant importance in the realm of sustainable company development [5,7].

The Regulation of the Minister of Tourism and Creative Economy/Head of the Tourism and Creative Economy Agency of the Republic of Indonesia Number 11 Year 2022 regarding The Strategic Plan of the Ministry of Tourism and Creative Economy/Tourism and Creative Economy Agency 2020–2024 outlines a total of 17 distinct creative sectors (Table 1). This expansion builds upon the initial 14 sectors that were previously identified. The establishment of the Ministry of Tourism and Creative Economy, along with its Creative Economy Agency (BEKFRAF), took place in 2015 with the aim of facilitating the growth and development of two distinct sectors: traditional cultural industries and new creative industries [5]. The stated objective of the organization is to effectively execute the Creative Economy plan and oversee its advancement across different locations inside the nation.

**Table 1.** Sub-sectors of creative industries in Indonesia [51].

| No | Sub-Sector |
| --- | --- |
| 1 | Advertising |
| 2 | Game developer |
| 3 | Architecture |
| 4 | Craft |
| 5 | Culinary |
| 6 | Fashion |
| 7 | Film, animation, and video |
| 8 | Interior design |
| 9 | Product design |
| 10 | Visual Communication Design |
| 11 | Music |
| 12 | Performance Art |
| 13 | Publishing |
| 14 | Television and radio |
| 15 | Fine arts |
| 16 | Photography |
| 17 | Mobile App |

Indonesia's diverse cultural heritage and dynamic creative industries offer a unique backdrop for the study of creative economy governance [52] and for emphasizing the role of cultural preservation and innovation [53]. Furthermore, the creative economy's contributions expand to regional development, shedding light on the spatial dynamics of creative industries in Indonesia [5,53]. Indonesia's creative economy is deeply intertwined with its diverse cultural heritage. Research by scholars has underscored the importance of cultural preservation in the creative sector [54]. Creative industries often draw upon traditional art forms, folklore, and local craftsmanship, contributing to the preservation and revitalization of Indonesia's rich cultural heritage [53].

The growth of the Indonesian creative economy is not without challenges. In October 2018, Indonesia, in its role as the MIKTA (Mexico, Indonesia, Republic of Korea, Türkiye, and Australia) Countries Coordinator, collaborated with other stakeholders to organize a supplementary event focused on harnessing the capabilities of the creative economy in order to accomplish the objectives outlined in the Sustainable Development Goals (SDGs). The event aimed to assess various challenges that could contribute to the advancement and growth of creative industries. These factors included the rise of the creative economy and creative networks as facilitators of sustainable development, the potential benefits and drawbacks associated with increased automation and digitization, the necessity of training, retraining, and upskilling for new industry positions, the role of creative industries in advocating for gender equality and diverse perspectives, the disparities in creative economic growth across different regions, the difficulties in sharing knowledge and practices between countries, the importance of integrating creative industries into national economic and development plans, and the significant potential for increased south-south cooperation to tap into untapped opportunities. Furthermore, the convergence of traditional and digi-

tal creative industries has created opportunities for innovation but also poses regulatory challenges [5,54].

### 1.4. Governance in the Creative Economy

Governments worldwide have recognized the creative economy's potential and have actively engaged in its promotion. Numerous studies have highlighted the diverse strategies employed by governments to support and stimulate creative industries. Such strategies encompass policy formulation, infrastructure development, investment in education and skills development, and the creation of cultural hubs [55,56]. The digital creative economy, characterized by its reliance on digital technologies and its emphasis on creativity, innovation, and intellectual property, represents a rapidly evolving and transformative sector. Effective governance is essential to balance innovation, entrepreneurship, and regulatory concerns within this dynamic landscape. The protection of intellectual property rights is also a central concern in the creative economy. Scholars have discussed the challenges of copyright enforcement, piracy, and the tension between fostering creativity and protecting creators' rights. Effective governance must strike a delicate balance between incentivizing innovation and safeguarding intellectual property [57,58].

Governance in the creative economy context is multifaceted, encompassing issues related to intellectual property protection, talent development, cultural heritage preservation, and fostering entrepreneurship. Scholars have discussed governance challenges in creative industries, emphasizing the tension between commercialization and cultural preservation [59,60]. These challenges present an intricate web of policy dilemmas that the Indonesian government must navigate, providing a fertile ground for the investigation of innovative governance mechanisms [61].

Despite the considerable potential of the creative economy, it continues to face obstacles, primarily stemming from the absence of adequate policies aimed at bolstering the industries. The progress of creative work may be hindered by inadequate funding, a lack of financial sustainability, and the absence of a standardized valuation system [61]. Thailand is confronted with the predicaments stemming from a dearth of consensus regarding the definition of creative industries and the swift transformations occurring in the digital realm [62,63]. Conversely, the Philippines necessitates the establishment of a centralized governmental entity to formulate a comprehensive policy framework for creative industries [62]. However, there is a dearth of research that investigates how the Indonesian government, in particular, adapts and innovates its governance mechanisms in the context of its unique cultural and economic landscape [64].

## 2. Materials and Methods

This research employs a qualitative methodology to explore the innovative governance strategies within the Indonesian digital creative economy from a governmental perspective. Qualitative research is well suited to capture the nuances, complexities, and contextual specifics of governance in this dynamic and evolving domain [65]. The qualitative approach allows for in-depth examination and interpretation of the experiences, perspectives, and actions of key governmental actors, and stakeholders involved in shaping the digital creative economy in Indonesia.

### 2.1. Data Collection

Data for this study were primarily collected through semi-structured interviews with a purposive sample of key informants, including government officials, policymakers, industry representatives, and experts in the field of the digital creative economy. The use of semi-structured interviews enables flexibility to explore emergent themes and gather rich, context-specific insights [66]. The interviews were conducted in-person and virtually, allowing for a diverse range of participants and geographic representation. In addition to interviews, document analysis was conducted to review relevant governmental policies, reports, legislation, and official documents related to the digital creative economy in In-

donesia. The document analysis provided valuable contextual information and corroborate findings from interviews.

### 2.2. Data Analysis

The collected qualitative data were thematically analyzed. It involved identifying, analyzing, and reporting patterns (themes) within the data [65]. This approach enabled the researcher to systematically examine and interpret the rich qualitative data, identifying key themes related to innovative governance strategies, challenges, opportunities, and policy implications within the Indonesian digital creative economy

### 2.3. Validity

Ensuring the trustworthiness and validity of qualitative findings is paramount [67]. To enhance the rigor of this study, strategies such as member checking, where participants reviewed and confirmed the accuracy of their interview transcripts, and peer debriefing, involving discussions with colleagues knowledgeable in qualitative research, were employed. Additionally, the use of triangulation, comparing findings from interviews with information from document analysis, enhanced the credibility and validity of the research [65].

### 2.4. Ethical Consideration

This research adhered to ethical guidelines and principles, including informed consent, confidentiality, and the protection of participants' identities. All participants were provided with clear information about the research objectives, their rights, and the handling of data.

### 3. Results

The participants in this research were mostly females (51.5%). Meanwhile, the rest (about 48.5%) were males, with a total of 103 government officials and civil servants; by age group, the majority (52.4%) of our respondents were 26 to 35 years old. Only 4.9% of respondents were aged between 46 to 55 years old. Variations in terms of the educational background were also captured from this survey. About 61.2% of civil servants in this survey had bachelor's degrees. Some of the respondents had a master's degree (26.2%) and only one person in this survey had a doctorate (1.0%). The rest had a variety of educational backgrounds, such as a high school degree or a technical/vocational diploma. With the variations in terms of age and educational background, most of the respondents had a tenure of working for about 2 to 5 years and could be categorized into level IVe.

Regarding the survey, the most responses were obtained from the Ministry of Cooperatives and SMEs (53.1%), where 70.59% of responses from the Ministry of Cooperatives and SMEs came from the Revolving Fund Management Institution (LPDB). While the second largest response was obtained through the Ministry of Tourism and Creative Economy (22.9%). The rest, 24%, were from other ministries. Looking at the volume of responses, we were heartened by the amount (53.1.9%) of individuals at or allied to the Ministry of Cooperatives and Small Medium Enterprise, especially the Revolving Fund Management Institution. This is an area of policymaking that can significantly contribute to the development of the creative economy. Also, the Ministry of Tourism and Creative Economy, along with the high level of understanding of creative economy, was the second largest response that we obtained (22.9%).

The "other" grouping was a truly diverse range of responses, arguably reflecting the reach and breadth of impact of the creative economy agenda. It included civil servants working in the Ministry of Trade, Foreign Affairs, Education and Culture, Industry, National Development Planning, and the Central Bank of Indonesia—all of which, as parts of government, are closely linked to the creative economy agenda and its remarkable ability to deliver beyond the perceived boundaries of art and culture. The survey respondents reflected the breadth of engagement by the Indonesian government in its creative economy agenda from core divisions in the Ministry of Tourism and Creative Economy that clearly utilize assets formed in the creative economy to deliver their distinct agenda. Consequently,

we believe that the data produced are an appropriate basis from which to draw conclusions that allow us to fulfill the objectives of this project.

## 4. Discussion

### 4.1. Understanding Creative Economy

In general, the knowledge and experience of respondents as seen from the percentage of time spent working on the Creative Economy agenda tended to be in the range of 26% to 75%. That is, the respondents in this study on average were involved in work on the creative economy within that range. With this, the specific sector became the majority of the areas of expertise/specialization found—in the order: (i) specific sector; (ii) social development; (iii) economic development; (iv) policy development; and (v) cultural development. The educational level also played a role in determining expertise/specialization in the creative economy. According to the data obtained, civil servants with an undergraduate education background had the most expertise in social development and specific sectors and only a few had expertise in policy development. Meanwhile, civil servants who had a master's degree educational background had the most expertise in policy development.

SMEs had a relationship with the creative economy. Most of the SME actors had businesses in the creative economy sub-sector. Although the majority of civil servants working in the Ministry of Cooperatives and SMEs believed that they only spent some time working on the creative economy agenda, less than 25%; sequentially, civil servants in this ministry claimed to have expertise in social development, specific sectors, economic development, cultural development, and, lastly, policy development. It is different with the Ministry of Tourism and Creative Economy, none of the civil servants working under this ministry had knowledge or experience in the creative economy field, at less than 25%. Civil servants in the Ministry tended to have expertise with a balanced number in each expertise listed—specific sectors, cultural development, economic development, policy development, and social development.

For responses in other ministries/agency categories, the majority of civil servants had spent time working on the creative economy agenda, in the range of 26% to 50%. Civil servants in this category felt they had expertise, respectively, in terms of (i) policy development, (ii) specific sectors, (iii) economic development, (iv) social development, and (v) cultural development. "Creative/creativity as a source of embodiment of added value from intellectual property in the law above" was found in every civil servant's answer when asked to describe the creative economy.

Of civil servants who had spent time working on the creative economy agenda, less than 25% and between 51% to 75%, believed that the definition of the creative economy had something to do with the word "knowledge". Civil servants who had spent time working on the creative economy agenda in the range of 26% to 50% believed that the definition of the creative economy was related to the words "ideas" and "intellectual property". Meanwhile, civil servants who had spent more than 75% of their time working on the creative economy believed that the definition of creative economy was related to the words "concept", "cooperatives", "intellectual property", and "technology". This shows that the respondents as a whole have been able to describe the creative economy in accordance with the definition by UNCTAD, as follows: "The creative economy has no single definition. It is an evolving concept that builds on the interplay between human creativity and ideas and intellectual property, knowledge and technology. Essentially it is the knowledge-based economic activities upon which the 'creative industries' are based. The creative economy is the sum of all the parts of the creative industries, including trade, labor and production. Today, the creative industries are among the most dynamic sectors in the world economy providing new opportunities for developing countries to leapfrog into emerging high-growth areas of the world economy".

Although an understanding of the definition was obtained, if there was training to develop understanding, then, of the civil servants who had spent time working on the creative economy agenda, less than 25% believed that seven out of sixteen training

practices were considered critical. Meanwhile, civil servants who had spent time working on the creative economy agenda in the range of 26% to 50% believed that intellectual property rights (particularly copyright) and strategies for creative hubs and clusters training consisted of training that should be focused on. It was different with civil servants who spent time working on the creative economy agenda in the range of 51% to 75%. They believed that technological impacts were the focus of the training needed. Lastly, civil servants who had spent time working on the creative economy agenda believed that economic impact was the focus of training to be considered critical.

*4.2. How an Organization Engages with Creative Economy*

In general, there are several interventions or policies needed to support creative economic growth. In Indonesia, initiatives that develop the skills and networks of creative entrepreneurs are the critical interventions or policies needed. In addition, (i) making techno-parks accessible to creative and cultural professionals and supporting joint projects, (ii) redrafting legislation on public procurements to better support creative and innovative initiatives, (iii) integrating the support for cultural and creative industries into education policies, and (iv) providing greater development and growth support for emerging micro and small creative business, is no less important to the support of the growth of the creative economy. However, (a) promoting the establishment of creative hubs and (b) promoting digital innovation investments in and by cultural and creative industries are considered unimportant for Indonesia's creative economy growth at this time.

Based on the operating level of intervention, there was no difference between civil servants who spent time working on the creative economy agenda—less than 25%, 26–50%, 51–75%, or above 75%. The majority of the interventions provided were still at the regional and national levels. It was similar for a number of ministries/agencies regarding the data obtained. The majority of the interventions provided were still at the regional and national level. Some then also carried out interventions at the city/municipality/town level.

A number of interventions, implemented at the operating scale above, have had an impact on the growth of the creative economy. Based on the time spent working on the creative economy agenda, creative impact (in terms of new approaches, new characters, new production) and economic impact (in terms of employment, investment, income, added value, and tourism) are the most expected impacts of the intervention that has already been made. To achieve this, regarding civil servants who spend time working on the creative economy agenda below 25%, universities, faculties, and research centers become promising partners. Meanwhile, for civil servants who spend time working on the creative economy agenda, 26% above, cultural and creative professionals and initiatives (including creative hubs and professional associations) are the main collaboration partners to work together in realizing good creative economic growth.

If the focus is on the relevant ministries, for the Ministries of Cooperatives and SMEs (also the Revolving Fund Management Institution/LPDB), the interventions or policies provided should have an impact on promotion and branding. Cultural and creative professionals and initiatives (including creative hubs) are related parties that should work together to realize good creative economic growth. Projects run with several partners tend to take 1 to 7 years—medium term. Civil servants at this particular ministry tend to be attracted by traditional cultural expressions (passively as consumers and as the audience) and digital interactive (actively as hobbyists/amateurs) sub-sectors.

It is different with the Ministry of Tourism and Creative Economy, the intervention or policy provided should have a key impact on the economic impact (in terms of employment, investment, income, added value, and tourism). Cultural and creative professionals and initiatives (including creative hubs) should be the related parties to work together to realize good creative economic growth. Projects run with several partners tend to take 1 to 7 years—medium term. Civil servants at this ministry tend to be attracted by performing arts, traditional cultural expressions, and cultural sites passively as consumers and audiences. They also tend to be actively attracted to the visual arts as a hobbyist/amateur.

For other ministries and agencies, the intervention or policy given should have a main impact on the economic impact (in terms of employment, investment, income, added value, and tourism). Cultural and creative professionals and initiatives (including creative hubs) are related parties that should work together to realize good creative economic growth. Projects run with several partners tend to take 1 to 7 years—medium term. Civil servants in several ministries and agencies tend to be attracted to digital interactive (as consumers and audiences) and traditional cultural expressions (actively as a hobbyists/amateurs).

*4.3. Collaboration*

Effective collaboration is one way to increase the growth of the creative economy. For civil servants who spent time working on the creative economy agenda, less than 25%, to create effective collaboration, skills in terms of responsibility and leadership are needed with strategic networks and knowledge-sharing approaches, which are types of approaches with high effectiveness. Increasing the effectiveness of collaboration requires tools that can be supported. For example, in establishing communication, WhatsApp is an application that is often used. Digital documentation of work tends to be stored in Google Drive. As for managing collaborative projects, Google, E-proposal, CMFS, Nitro, and Adobe are used. Generally, collaborations carried out take place with a fairly rare frequency—three to four times a year, or have happened in the past, but there has been no further collaboration. For that, they need to be able to establish a wider collaboration.

Meanwhile, of the civil servants who spent time working on the creative economy agenda, between 26% and 50%, believed that communication and problem-solving were the skills needed to collaborate through high-effectiveness approaches—share projects, knowledge sharing, and catch-ups. To support this, several tools are used, such as WhatsApp, Google, and Zoom (as platforms for teams and communication). In addition, for the purposes of digital documents and digital project management, Google Drive and Google are most often used. Generally, collaborations carried out by civil servants take place at a fairly frequent frequency—monthly or annually. However, it is possible that there are still people who are not aware of any collaboration. Therefore, various problems are still faced at this level, such as organization, policy, and regulatory.

For civil servants who spent time working on the creative economy agenda, between 51% to 75%, resource management and problem-solving were skills needed for effective collaboration with a strategic network collaboration approach. To support this, several tools are used, such as Instagram, Teams, and Zoom (as platforms for teams). Communication is often conducted through WhatsApp and Telegram. In addition, for digital document purposes, Google Drive is the most frequently used platform. Generally, collaborations are carried out with varied frequency—three to four times a year or annually. For that, they must have the desire to be able to establish a wider collaboration. However, there are several obstacles experienced in establishing effective collaboration, mostly in communication.

For civil servants who spent time working on the creative economy agenda, more than 75%, believed that a positive attitude and commitment to the collaboration process were needed for effective collaboration with a cross-departmental team approach, feedback sessions, strategic networks, and knowledge sharing. To support this, several tools are used, such as E-mail, Google Meet, WhatsApp, and Zoom (as a platform for teams and communication). In addition, for the purposes of digital documents, Google Drive is the most frequently used platform. Generally, collaborations carried out take place with varied frequency—monthly, three to four times a year, or have happened in the past, but there has been no further collaboration. For that, they have the desire to be able to establish a wider collaboration. However, there have been several obstacles experienced in establishing effective collaboration, including cooperation and projects.

*4.4. Job Satisfaction and Organizational*

In every job, it is always possible that there will be obstacles in working on projects, collaboration, or other things. However, despite all the challenges that exist, it is important

that employees feel happy and engaged with their work. One of the factors that influence this is pay grade. Based on the group, the pay grade of civil servants could be categorized into 17 groups ranging from Ia to IVe. The variety of pay grades based on these groups did not significantly affect the level of happiness and engagement of civil servants with their work. Both groups Ia to IVe felt happy and engaged in their work evenly—from: (i) opportunity to develop a career; (ii) sense of personal achievement; (iii) all challenges; to (iv) interest. However, the pay grades of groups IIIa, IIIb, IIId, and IVa felt happy and engaged in their work because their work was interesting.

Whether it was through the time spent working on the creative economy agenda or from the ministry where civil servants worked, they all agreed that they take great pride in telling people where they work. In addition, civil servants working in the Ministry of Cooperatives and SMEs described that they had strong personal attachments to their organizations and that senior managers were quick to respond. Therefore, most of them did not think about resigning/leaving the organization. However, for some civil servants who were thinking of resigning/leaving the organization, most of them had not decided yet whether to continue to work on the creative economy agenda or not.

Wherever civil servants work, regardless of their pay grade, the opportunity to gain learning and development is something that must be sought and felt. Through time spent working on the creative economy agenda, learning and development opportunities are obtained from access and Continuing Professional Development/CPD (only for time spent working on the creative economy agenda, around 26% to 50%). The forms of learning and development are also quite diverse. For those who spent time working on the creative economy agenda, less than 25%, placements elsewhere in government, in external agencies, or in a creative business, as well as similar opportunities, would help them to develop an understanding of how the government can best support creative business in their development. As for those who spent time working on the creative economy agenda in addition to that, they preferred to keep up to date with changes and also read and research. If reviewed through each relevant ministry, access was the main key to obtaining learning and development opportunities. For the Ministry of Tourism and Creative Economy and the Ministry of Cooperatives and SMEs, they preferred to keep up to date with changes, also reading and research, as a form of learning and development. For other ministries/agencies, placements elsewhere in government, in external agencies, or in a creative business, and similar opportunities, would help them to develop understanding of how the government could best support creative businesses in their development.

Encouraging innovation is something that will continue and be carried out, especially when it involves the creative economy. This often involves teamwork that can be built between civil servants. According to the civil servant, the team where they work always focuses on the delivery of benefits on creative business that can be achieved, mainly by (i) having a clear understanding of the organization's objectives and (ii) understanding the role of the organization in a wider setting. With this, the organization values creativity and innovation within the organizational culture.

Readers should discuss the results and how they can be interpreted from the perspective of previous studies and the working hypotheses. The findings and their implications should be discussed in the broadest context possible. Future research directions should also be highlighted.

## 5. Conclusions

To improve the understanding of the creative economy, training related to sub-sectors and regular knowledge sharing from industry experts or seminars related to the creative industry are needed to increase the level of expertise of civil servants (currently there are 29.1% who do not have expertise—broad or development expertise). The churn rate is still high. According to LinkedIn, the average annual worldwide employee turnover rate is 10.9%. The ministry needs to think about how to decrease this number. Some efforts to be considered are:

- Find the right talent;
- Encourage retention early on;
- Recognize and reward employees;
- Identify a clear career path;
- Encourage a healthy work-life balance; and
- Create learning and development programs.

Effective collaboration is seen as a vital part of ensuring that any organization effectively performs. To undertake this collaboration, we suggest that the government creates a digital innovation co-creation platform for civil servants (Figure 1). The platform itself should aim for:

- Open/better communication: communication tools are clearly already a common thing to have, but skills, check-in projects, and evaluation check-ins are some things to be considered.
- To anticipate different alignments of visions and objectives, a project charter, and a matrix that are agreed on by all parties for each project.
- To manage the availability of resources (or lack of it), collaboration between departments between ministries, or involving other stakeholders.

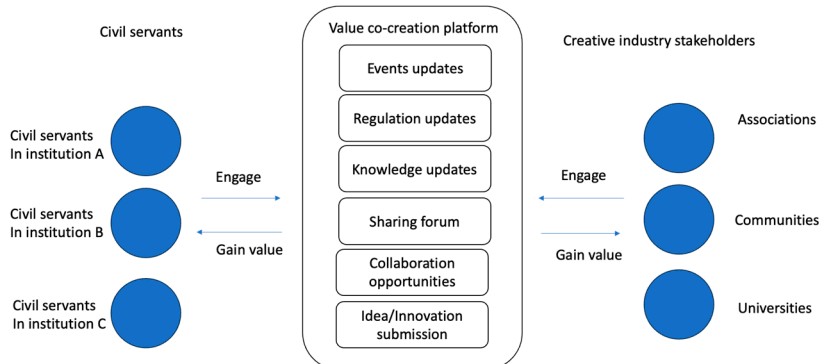

**Figure 1.** Digital Innovation Co-Creation Scheme for Civil Servants.

A strategic roadmap is essential for the creation and implementation of a digital transformation platform that fosters co-creation among stakeholders. To develop such a platform, a government can follow these steps:

First, the government should conduct stakeholder analysis and engagement. At this stage, the government can decide which groups will be involved in the co-creation process and engage with them through workshops, surveys, and forums to understand their needs, expectations, and challenges. Secondly, the government must define goals and objectives. Start by establishing a clear vision for what the co-creation platform aims to achieve in terms of digital transformation, then set specific, measurable, achievable, relevant, and time-bound (SMART) objectives for the platform. Thirdly, platform design and development are needed. The design of the platform should have a strong focus on user experience, ensuring it is accessible, intuitive, and responsive. The government should choose the right technology stack that is scalable, secure, and can integrate with existing government systems. Fourth, the pilot programs should be carried out with a select group of stakeholders to test the platform's functionality and to gather initial feedback. The feedback should then be used to make improvements before a full-scale rollout. Fifth, the government should ensure that the platform complies with all relevant legal, privacy, and security requirements. Collaboration and the sharing of data and resources should be encouraged by policies. For stakeholders to effectively use the platform, capacity building and training should be provided. Additionally, a helpdesk or support team to assist users and encourage adoption is needed. Seventh, conduct a platform launch for a broader audience for additional testing and feedback. Then, officially roll out the platform with a marketing campaign to encourage engagement. Eighth, make sure there is

a collaboration mechanism in the platform. This could include communication tools such as messaging, forums, and video conferencing to facilitate communication, as well as idea management features for submitting, voting on, and discussing ideas. Then, make sure monitoring, evaluation, and iterative improvement are available. The key performance indicators (KPIs) and metrics to monitor the platform's usage and the progress of digital transformation initiatives are needed, along with user feedback that is regularly collected to understand how the platform can be improved. To conclude, the government should develop a sustainable financial model, possibly with public-private partnerships, and plan for the scaling of the platform in terms of technology and stakeholder participation.

**Author Contributions:** Conceptualization, D.D. and N.A.; methodology, D.D. and T.R.F.; software, T.R.F.; validation, D.D. and N.A.; formal analysis, N.A.; investigation, D.D. and N.A.; resources, D.D.; data curation, N.A.; writing—original draft preparation, D.D. and T.R.F.; writing—review and editing, D.D. and N.A.; visualization, D.D.; supervision, D.D.; project administration, D.D.; funding acquisition, N.A. All authors have read and agreed to the published version of the manuscript.

**Funding:** This research received no external funding.

**Institutional Review Board Statement:** Not applicable.

**Informed Consent Statement:** Informed consent was obtained from all subjects involved in the study.

**Data Availability Statement:** No new data were created or analyzed in this study. Data sharing is not applicable to this article.

**Conflicts of Interest:** The authors declare no conflict of interest.

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
