# Peer review of "Digital Innovative Governance of the Indonesian Creative Economy: A Governmental Perspective"

_sustainability, doi:10.3390/su152316234_

Round 1

Reviewer 1 Report

Comments and Suggestions for Authors

Firstly, in my opinion, the relevance of the study is not strictly substantiated. In particular, it is hardly a serious argument for an international journal that the processes of development of the creative economy in Indonesia have been little studied. However, it is not indicated why the Indonesian case is interesting and how it differs from other countries.

Secondly, the article talks a lot about the respondents - government officials from Indonesia, as well as how immersed they are in the topic of the creative economy and using what tools they interact with each other. However, nothing is said about how this affects the development of the creative economy in Indonesia. If this is just planned, then publication of the article in an international journal is still premature.

In general, I lacked information about how the creative economy processes work in Indonesia, what their features are, where they experience difficulties, how the Indonesian government helps overcome these difficulties, and, as a result, what Indonesian officials lack in order to successfully solve problems related to the development of the creative economy.

Author Response

Response to Reviewer 1 Comments

Summary

Thank you very much for reviewing this manuscript. The detailed responses are below, along with the corresponding revisions/corrections highlighted in the resubmitted files.

Point-by-point response to Comments and Suggestions for Authors

Comments 1: Firstly, in my opinion, the relevance of the study is not strictly substantiated. In particular, it is hardly a serious argument for an international journal that the processes of development of the creative economy in Indonesia have been little studied. However, it is not indicated why the Indonesian case is interesting and how it differs from other countries.

Response 1:

Agree. We have, accordingly, add explanation in Line 150-161 to emphasize this point. I hereby attached the changes for comments 1 below.

Indonesia has consistently advocated for the development and promotion of the creative economy. In 2018, Indonesia provided support for the Bali Agenda for Creative Economy, and also witnessed the United Nations General Assembly's approval of a resolution designating 2021 as the Year of the Creative Economy for Sustainable Development. As the designated host country for the G20 summit in 2022 and the chair of the ASEAN secretariat in 2023, our objective is to enhance coordination and collaboration at the local, regional, and global levels to bolster the creative economy as a catalyst for content-oriented economic growth. The creative economy serves as a valuable resource for facilitating the availability of employment opportunities, particularly within micro, small, and medium-sized firms, during periods characterized by swift transformations in labor markets. The creative economy holds significant importance in the realm of sustainable company development [5][7].

Comments 2 & 3: Secondly, the article talks a lot about the respondents - government officials from Indonesia, as well as how immersed they are in the topic of the creative economy and using what tools they interact with each other. However, nothing is said about how this affects the development of the creative economy in Indonesia. If this is just planned, then publication of the article in an international journal is still premature.

In general, I lacked information about how the creative economy processes work in Indonesia, what their features are, where they experience difficulties, how the Indonesian government helps overcome these difficulties, and, as a result, what Indonesian officials lack in order to successfully solve problems related to the development of the creative economy.

Response 2 & 3: Agree. We have, accordingly, add explanation in Line 183-199 to emphasize this point. I hereby attached the changes for comments 2& 3 below.

The growth of the Indonesian creative economy is not without challenges. In October 2018, Indonesia, in its role as the MIKTA (Mexico, Indonesia, South Korea, Turkiye, and Australia)  Countries Coordinator, collaborated with other stakeholders to organize a supplementary event focused on harnessing the capabilities of the creative economy in order to accomplish the objectives outlined in the Sustainable Development Goals (SDGs). The event aimed to assess various challenges that could contribute to the advancement of the creative industries' growth. These factors included the rise of the creative economy and creative networks as facilitators of sustainable development, the potential benefits and drawbacks associated with increased automation and digitization, the necessity of training, retraining, and upskilling for new industry positions, the role of creative industries in advocating for gender equality and diverse perspectives, the disparities in creative economic growth across different regions, the difficulties in sharing knowledge and practices between countries, the importance of integrating creative industries into national economic and development plans, and the significant potential for increased South-South cooperation to tap into untapped opportunities.  Furthermore, the convergence of traditional and digital creative industries has created opportunities for innovation but also poses regulatory challenges [5][54].

Reviewer 2 Report

Comments and Suggestions for Authors

The rapidly unfolding digital transformation of the creative economy presents countries with both unprecedented challenges and unparalleled opportunities. This paper takes a deep dive into Indonesia's journey, attempting to balance innovation with regulation. By employing qualitative methodologies to understand the measures taken by the Indonesian government, the research delivers a detailed overview of the governance mechanisms in place to nurture, regulate, and optimize the digital creative economy.

A standout element of this paper is its examination of real-world experiences and perspectives from government officials, policymakers, and stakeholders. The utilization of semi-structured interviews and document analysis paints a holistic picture of how the Indonesian civil service, policies, and the creative economy intermingle, especially in a developing nation context. The findings offer a wealth of knowledge, from adaptive strategies to crucial policy implications, elucidating the intricate dance between governance and the digital age of creative industries.

The paper delves deep into pressing issues faced by the creative economy. One of the notable areas highlighted is the challenge of upskilling civil servants in creative economy nuances. With almost a third lacking broad or developmental expertise, there is a clear and pressing need for dedicated training sessions, regular knowledge-sharing endeavors, and seminars that tap into industry expertise.

Furthermore, the high turnover rate among employees, which surpasses the global average as per LinkedIn data, underscores a pressing concern for Indonesia. The paper astutely suggests practical solutions to curb this trend – from hiring the right talent to promoting a balanced work-life culture. Each recommendation is comprehensive, aiming to foster a conducive environment for professionals within the creative industry.

A significant contribution of this paper is the emphasis on effective collaboration. As the cornerstone of any organization's success, the proposed digital innovation co-creation platform for civil servants could be a game-changer. By addressing concerns like open communication, alignment of visions, and effective resource management, this platform could provide a robust structure for departments and stakeholders to seamlessly interact.

To further enhance the impact of this research, I recommend the following extensions:

->Regional Comparisons: Examining how other nations in Southeast Asia, with similar socio-economic and digital landscapes, address similar challenges would offer comparative insights beneficial for policy refinements.

->Feedback from Ground Level: Including the experiences and suggestions of grassroots-level workers in the creative economy could provide a more layered understanding of the digital transformation's effects.

->Implementation Challenges: A deep dive into potential challenges in implementing the recommended strategies and a roadmap to overcome these would be invaluable.

In conclusion, this paper provides an incisive look into Indonesia's efforts to marry governance with the burgeoning digital creative economy. Its findings and recommendations are not just pertinent to Indonesia but have broader implications for nations globally. With the suggested extensions, the research could offer even more profound insights into navigating the complexities of the digital age in the creative sector.

Author Response

Response to Reviewer 2 Comments

Summary

Thank you very much for reviewing this manuscript. The detailed responses are below, along with the corresponding revisions/corrections highlighted in the resubmitted files.

Point-by-point response to Comments and Suggestions for Authors

Comments 1: Regional Comparisons: Examining how other nations in Southeast Asia, with similar socio-economic and digital landscapes, address similar challenges would offer comparative insights beneficial for policy refinements.

Response 1:

Agree. We have, accordingly, add explanation in Line 222-232 to emphasize this point. I hereby attached the changes for comments 1 below.

Despite the considerable potential of the creative economy, it continues to face obstacles, primarily stemming from the absence of adequate policies aimed at bolstering the industries. The progress of creative work may be hindered by inadequate funding, a lack of financial sustainability, and the absence of a standardized valuation system [66]. Thailand is confronted with the predicaments stemming from a dearth of consensus regarding the definition of creative industries and the swift transformations occurring in the digital realm [67]. Conversely, the Philippines necessitates the establishment of a centralized governmental entity to formulate a comprehensive policy framework for creative industries [66]. However, there is a dearth of research that investigates how the Indonesian government, in particular, adapts and innovates its governance mechanisms in the context of its unique cultural and economic landscape [62].

Comments 2 & 3: Feedback from Ground Level: Including the experiences and suggestions of grassroots-level workers in the creative economy could provide a more layered understanding of the digital transformation's effects

Implementation Challenges: A deep dive into potential challenges in implementing the recommended strategies and a roadmap to overcome these would be invaluable

Response 2 & 3: Agree. We have, accordingly, add explanation in Line 538-573 to emphasize this point. I hereby attached the changes for comments 2& 3 below.

A strategic roadmap is essential for the creation and implementation of a digital transformation platform that fosters co-creation among stakeholders. To develop such a platform, a government can follow these steps:
First, the government should conduct a stakeholder Analysis and Engagement. At this stage, the government can decide which groups will be involved in the co-creation process and engage them through workshops, surveys, and forums to understand their needs, expectations, and challenges. Secondly, the government must define Goals and Objectives. Start by establishing a clear vision for what the co-creation platform aims to achieve in terms of digital transformation. then set specific, measurable, achievable, relevant, and time-bound (SMART) objectives for the platform. Thirdly, platform design and development is needed. The design of the platform should have a strong focus on user experience, ensuring it is accessible, intuitive, and responsive. The government should choose the right technology stack that is scalable, secure, and can integrate with existing government systems.  Fourth, the pilot programs should be carried out with a select group of stakeholders to test the platform’s functionality and to gather initial feedback. The feedback then used to make improvements before a full-scale rollout.  Fifth, the government should ensure that the platform complies with all relevant legal, privacy, and security requirements. Collaboration and sharing of data and resources should be encouraged by policies. For stakeholders to use the platform effectively, capacity building and training should be provided. Additionally, a helpdesks or support teams to assist users and encourage adoption is needed. Seventh, conduct a platform launch to a broader audience for additional testing and feedback. then later roll out the platform officially with a marketing campaign to encourage engagement. Eight, make sure there is a collaboration mechanism in the platform. This could include communication tools such as messaging, forums, and video conferencing to facilitate communication. and idea management features for submitting, voting on, and discussing ideas. Nineth, make sure the availability of monitoring, evaluation, and iterative improvement. The key performance indicators (KPIs) and metrics to monitor the platform’s usage and the progress of digital transformation initiatives is needed along with the user feedback that is collected regularly to understand how the platform can be improved. To conclude, the government should develop a sustainable financial model, possibly with public-private partnerships, and plan for the scaling of the platform in terms of technology and stakeholder participation.

Round 2

Reviewer 1 Report

Comments and Suggestions for Authors

-

Reviewer 2 Report

Comments and Suggestions for Authors

Paper can be accepted now